# Containment of *Fusarium culmorum* and Its Mycotoxins in Various Biological Systems by Antagonistic *Trichoderma* and *Clonostachys* Strains

**DOI:** 10.3390/jof9030289

**Published:** 2023-02-22

**Authors:** Lidia Błaszczyk, Hanna Ćwiek-Kupczyńska, Karolina Hoppe Gromadzka, Aneta Basińska-Barczak, Łukasz Stępień, Joanna Kaczmarek, Leszek Lenc

**Affiliations:** 1Institute of Plant Genetics, Polish Academy of Sciences, Strzeszyńska 34, 60-479 Poznań, Poland; 2Department of Chemistry, Faculty of Forestry and Wood Technology, Poznan University of Life Sciences, Wojska Polskiego 75, 60-625 Poznań, Poland; 3Department of Phytopathology and Molecular Mycology, University of Technology and Life Sciences, Prof. S. Kaliskiego 7 Street, 85-796 Bydgoszcz, Poland

**Keywords:** *Trichoderma atroviride*, *Trichoderma cremeum*, *Trichoderma viride*, *Clonostachys rosea*, solid substrate and liquid bioassay, HPLC-MS, light and scanning electron microscopy, quantitative PCR, *Triticum aestivum* L., wheat yield parameters

## Abstract

Prevention of fungal diseases caused by *Fusarium* species, including *F. culmorum*, and thus the accumulation of mycotoxins in wheat ears, is a constant challenge focused on the development of new, effective crop management solutions. One of the currently most ecologically attractive approaches is biological control using natural antagonistic microorganisms. With this in mind, the antagonistic potential of thirty-three *Clonostachys* and *Trichoderma* strains was assessed in this work. Screening tests were carried out in in vitro cultures, and the observed potential of selected *Trichoderma* and *Clonostachys* strains was verified in field and semi-field experiments with two forms of wheat: winter cv. Legenda and spring cv. Bombona. Three strains, namely *C. rosea* AN291, *T. atroviride* AN240 and *T. viride* AN430 were reported to be most effective in inhibiting the growth of *F. culmorum* KF846 and the synthesis of DON, 3AcDON and ZEN under both laboratory and semi-controlled field conditions. Observations of the contact zones of the tested fungi in dual cultures exposed their mycoparasitic abilities against KF846. In addition, studies on liquid cultures have demonstrated the ability of these strains to eliminate *F. culmorum* toxins. Meanwhile, the strains of *T. atroviride* AN35 and *T. cremeum* AN392 used as soil inoculants in the field experiment showed a different effect on the content of toxins in ears (grains and chaffs), while improved wheat yield parameters, mainly grain health in both wheat cultivars. It is concluded that the selected *Trichoderma* and *Clonostachys* strains have a high potential to reduce the adverse effects of *F. culmorum* ear infection; therefore, they can be further considered in the context of potential biocontrol factors and as wheat crop improvers.

## 1. Introduction

*Fusarium culmorum* (W.G. Smith) Sacc. is an important pathogen of small kernel cereals worldwide [1,2]. Together with *F. graminearum* Schwabe (teleomorph *Gibberella zeae*), it has been reported as one of the predominant species causing Fusarium Foot and Root Rot (FRR) and Fusarium Head Blight (FHB) of wheat [1]. *Fusarium culmorum* is frequently isolated in the temperate or cooler regions in Europe and Canada [3]. However, in the last twenty years, this pathogen was also reported as the causative agent of FHB in Italy, Spain, Turkey, Tunisia [4,5,6], Australia and New Zealand [7]. These incidents were observed in the years with climatic conditions that favor the development of the pathogen, characterized by wet weather during the phenological phases of flowering and milk stage [8].

The significance of *F. culmorum* in wheat production is mainly related to the infection of wheat heads by this pathogen, followed by fungal colonization of the ear and the contamination of kernel with unacceptable, high amounts of mycotoxins, which are a potential health hazard for both humans and animals [9]. *Fusarium culmorum* is capable of producing two major types of mycotoxins: type B trichothecenes, such as deoxynivalenol (DON), 3-acetyl-deoxynivalenol (3-AcDON), 15-acetyldeoxynivalenol (15-ADON), nivalenol (NIV) and 4-acetylnivalenol (4-ANIV, syn. fusarenone-X) and estrogenic zearalenone (ZEN) [3,10]. Depending on trichothecene production, two chemotypes have been distinguished in *F. culmorum*: chemotype I, which produces deoxynivalenol and/or its acetylated derivatives, and chemotype II, which produces nivalenol and 4-acetylnivalenol [10,11]. The sesquiterpene epoxide trichothecenes are considered to be the most bioactive compounds produced by *F. culmorum*. These metabolites have a pronounced cytotoxic effect on eukaryotic organisms [12]. Their high phytotoxicity has also been documented [13,14,15]. The mechanism of trichothecenes action is based primarily on the inhibition of protein synthesis and oxidative damage to cells, which in turn leads to the interruption of nucleic acid synthesis and finally to apoptosis [11,16]. Another mycotoxin produced by *F. culmorum* is zearalenone. In terms of chemical structure, this metabolite is an analogue of naturally occurring estrogens, including estradiol, estrone, estriol, 7-β-estradiol or 17-β-estradiol [17]. As a xenoestrogen, it has an affinity for estrogen receptors, which leads to disorders in the endocrine regulation of reproductive functions in farm animals and may even cause hyperestrogen syndromes in humans [18,19]. Zearalenone has also been shown to be hepatotoxic, hematotoxic, immunotoxic and genotoxic, although the precise mechanism of ZEN toxicity has not been fully established [18,20]

With the above risks in mind, it is of great interest to find ways to reduce the impact of FHB caused by *F. culmorum* and to prevent mycotoxins from entering the food chain. Two approaches, prevention and intervention, are commonly used in the control of FHB. Prevention includes crop rotation with non-host species, management of remaining crop residues through tillage and resistance breeding [21]. Interventions include the use of crop ear protection measures, mainly fungicides, and management strategies during or after harvest (e.g., kernel treatment) [21]. Recently, more and more attention has been paid to natural plant protection products, mostly due to the need of reducing the negative impact of synthetic pesticides on the environment and emerging social concerns related to the health and safety of cereal products [22]. The search for new, health-promoting and ecological solutions in the fight against cereal diseases has resulted from several simultaneous legal actions of such organizational units as the Food and Agriculture Organization (FAO), the World Health Organization (WHO), the International Agency for Research on Cancer (IARC) and the European Union (EU). These actions concern the introduction of specific maximum limits for mycotoxins in food, including cereal-based products [23] and the sustainable use of pesticides and the promotion of the use of integrated pest management (IPM), including non-chemical alternatives to pesticides [24]. Recently, a new regulation on the sustainable use of plant protection products was proposed, including EU-wide targets to reduce by 50% the use and risk of chemical pesticides by 2030, in line with the EU’s “Farm to Fork” and “Biodiversity” strategies [25]. The above legal bases allow the development of one of the most promising strategies for managing the diseases caused by *F. culmorum*, which is biological control using beneficial, pathogen-antagonistic microorganisms. Among the microorganisms known for their natural potential to attack other fungi through competition for nutrients and/or space, antibiosis associated with the secretion of metabolites and direct mycoparasitism consisting in the production of lytic enzymes that degrade cell walls, there are fungi of the genus *Trichoderma* and *Clonostachys* [26,27]. It has been repeatedly documented that some species of these genera have the ability to reduce the population of pathogens in a variety of agricultural conditions and protect plants against diseases caused by pathogenic fungi, while improving vegetative growth and crop yield [26,27].

Therefore, the main goal of this study was to assess the potential of *Clonostachys rosea*, *Trichoderma atroviride*, *Trichoderma citrinoviride*, *Trichoderma cremeum*, *Trichoderma hamatum*, *Trichoderma harzianum*, *Trichoderma koningiopsis*, *Trichoderma longibrachiatum*, *Trichoderma longipile*, *Trichoderma viride*, and *Trichoderma viridescens* species to inhibit the growth of *F. culmorum* KF846 and to limit the production and content of zearalenone, deoxynivalenol and 3-acetyl-deoxynivalenol in culture media under laboratory conditions and in wheat kernels grown under field conditions. This characterization will serve the selection of potential broad-spectrum antagonists against the toxigenic *F. culmorum*.

## 2. Materials and Methods

### 2.1. Fungal Isolates

Thirty-two strains from the order of *Hypocreales*—one *F. culmorum*, six *C. rosea* and twenty-five *Trichoderma* strains were chosen for this study from the culture collection of the Institute of Plant Genetics, Polish Academy of Science, Poznań, Poland (Appendix A). All the studied *Trichoderma* strains, originating from different ecological niches in Poland, were identified to the species level by sequence analysis of the internal transcribed spacer regions ITS-1 and ITS-2 of the nuclear rDNA and a fragment of the translation elongation factor gene *tef1α* during earlier studies of Błaszczyk et al. [28,29,30]. *Clonostachys* strains were isolated from decaying wood, beans, soybeans, leafs and soil and were identified as described by Błaszczyk et al. [30] and Gromadzka et al. [31]. *Fusarium culmorum* KF 846 strain has been widely used in resistance screening experiments in wheat breeding [32,33]. The species identity and chemotype were already confirmed using gene-specific molecular markers [34] and analysis of the ITS and t*ef1α* sequence as described by Błaszczyk et al. [30] and Gromadzka et al. [31].

### 2.2. Dual Culture Bioassay

#### 2.2.1. Potato Dextrose Agar (PDA) Dual Culture Bioassay

Antagonistic potential of all *Trichoderma* and *Clonostachys* strains towards *F. culmorum* KF846 was evaluated in dual cultures on potato dextrose agar (PDA, Sigma Aldrich, Darmstadt, Germany) medium. Preserved strains of *Trichoderma*, *Clonostachys* and *Fusarium* were revived through the re-culturing from saltwater nutrient agar (SNA) [35] stock slants onto 8.5 cm diameter Petri dishes containing 15 mL of PDA medium. The plates were incubated for 7 days at 25 ± 2 °C subsequently. Mycelial discs of 3 mm-diameter from the last daily gain were used as inoculum in a *F. culmorum* KF846/ *Trichoderma* or *Clonostachys* strain combination. Mycelial discs of the pathogen and potential antagonist were placed on a PDA medium in 8.5 cm diameter Petri dishes at a distance of 8 cm from each other and were incubated at 25 ± 2 °C through 12 h/12 h of darkness/light. Individual cultures of the pathogen and potential antagonist were used as controls. Each culture/dual culture was completed in triplicate. The radial growth of each fungus was measured with a ruler daily for 7 days. The inhibition of mycelial growth and the qualitative evaluation of antagonism was carried out as described by Błaszczyk et al. [36] according to the formula: (Rc − R)/ Rc × 100, where Rc, R are the estimates of radial growth of a pathogen in the control and dual culture, respectively [37]. The final assessment of inhibitory effectiveness of the antagonists was based on a visual assessment of the colonies at day 14 of co-incubation. The most and the least efficient antagonists were identified based on a nominal scale (modified scale of Mańka [38]).

#### 2.2.2. Solid Substrate Bioassay

The potential of *Trichoderma* and *Clonostachys* to reduce the production of mycotoxins by the strain *F. culmorum* KF846 was determined based on the dual cultures on rice kernels according to the procedures described by Błaszczyk et al. [36]. Three *Trichoderma* (*T. atroviride* AN240, *T. koningiopsis* AN251, *T. viride* AN430) and two *C. rosea* (AN291, AN298) strains characterized by the highest antagonistic potential against *F. culmorum* KF846 in dual cultures on PDA, were selected for this analysis. The rice kernels (50 g) prepared in 300 mL Erlenmeyer flasks as described by Błaszczyk et al. [36] were inoculated with four disks (3 mm in diameter) of 7-day old PDA cultures of pathogen and/or potential antagonist. Individual cultures of *F. culmorum* KF846, *T. atroviride* AN240, *T. viride* AN430, *C. rosea* AN291 and AN298 and un-inoculated rice were used as controls. Each culture/ dual culture/non-culture was completed in triplicate. The flasks were incubated at 25 ± 2 °C 12 h/12 h darkness/light for 21 days. The samples were shaken daily to prevent clumping and to aerate the culture. Relative toxin reduction (%) was calculated based on the estimated toxin production level for the control and dual culture, according to a formula: RTR = (Tc − T)/ Tc × 100, where Tc, T are their respective toxin levels [36].

#### 2.2.3. Liquid Substrate Bioassay

To investigate the potential of *Trichoderma* and *Clonostachys* to remove exogenous mycotoxins ZEN, DON and 3-AcDON (Sigma-Aldrich, Steinheim, Germany) from liquid medium, the selected *T. atroviride* AN240, *T. viride* AN430 and *C. rosea* AN291 strains were passaged in axenic conditions from 7-day old PDA culture (three discs, 3 mm in diameter) to 100 mL Erlenmeyer flasks with 20 mL Czapek—Dox broth medium (Sigma-Aldrich, Saint Louis, MI, USA) supplemented with yeast extract (10 g/L, Oxoid™ Yeast Extract Powder Thermo Fisher Scientific, Waltham, MA, USA). The flasks were incubated at 25 °C and 200 rpm in a rotary shaker. After 48 h of incubation a mycotoxin dissolved in methanol (HPLC grade, Sigma-Aldrich, Steinheim, Germany) and used as stock solution was added to each culture: zearalenone–0.5 ng/μL, deoxynivalenol—10 ng/μL, 3-acetylodeoxynivalenol—0.5 ng/μL). As controls, flasks containing medium without a mycotoxin (matrix control) or without biomass (substrate control) were incubated in parallel. At zero time and at 1, 2, 4, 8, 24, 48 and 96 h, 1 mL of the culture was collected and centrifuged (12,000× *g* for 10 min). The supernatants were stored frozen (−20 °C) until the analysis. Experiments were repeated at least three times.

### 2.3. Model Semi-Field and Field Experiments

#### 2.3.1. Plant Material

The plant materials were two bread wheat cultivars: spring wheat cultivar Bombona, developed by Danko Hodowla Roślin Sp. z o.o z/s w Choryni (Kościan, Polska). and winter wheat cultivar Legenda, developed by Poznań Plant Breeders Ltd. (Tulce, Poland). These are common cultivars in Poland and are considered to be moderately resistant to fungal diseases according to the agronomic assessments made by the owner companies.

#### 2.3.2. Field and Semi-Field Experiment Designs

The field trials were conducted in 2017/2018 in Cerekwica (Western Poland; GPS coordinates: 52°5′ N, 16°7′ E)—an experimental area of the Institute of Plant Genetics of the Polish Academy of Sciences characterized by poor sandy-clay soil. The experiments were performed to determine the effects of the selected *T. atrovirde* AN35 and *T. cremeum* AN392 strains on the yield parameters and mycotoxins contamination in spikes of wheat plants. *Trichoderma atroviride* AN35 and *T. cremeum* AN392 were previously found to be able to colonize wheat roots and their internal tissues [39]. Semi-field studies were carried out in Lisewo Malborskie (Northern Poland, GPS coordinates: 54°6′ N, 18°43′ E), on brown soil of class I quality, in a conventional system, without the use of chemical plant protection products. In order to determine the effect of the pre-inoculation of selected *Trichoderma* (AN240, AN430) and *Clonostachys* (AN291) strains on the contamination of wheat with *F. culmorum* KF846 mycotoxins, semi-field studies were carried out under covers ensuring moisture requirements for fungal development.

The experiments were carried out according to the same scheme. The experimental field was divided into 9 (field trials) or 24 (semi-field trials under covers) replicate plots for each wheat cultivar. The size of each plot was 2 m^2^ with spaces of 10 cm between rows and 60 cm between plots or 1 m^2^ with spaces of 10 cm between rows and 30 cm between plots, respectively. A total of 50 g of kernels were used to sow one plot. A completely randomized design was used with three replicates of three (field trials) treatments, which included: (1) sterile distilled water—control; (2) *T. atroviride* AN35 inoculum; (3) *T. cremeum* AN392 inoculum; or eight (semi-field trials) treatments, which included: (1) sterile distilled water—negative control; (2) *T. atroviride* AN240 inoculum—positive control; (3) *T. viride* AN430 inoculum—positive control; (4) *C. rosea* AN291 inoculum—positive control; (5) *F. culmorum* KF846 inoculum—positive control; (6) *T. atroviride* AN240 inoculum + *F. culmorum* KF846 inoculum; (7) *T. viride* AN430 inoculum + *F. culmorum* KF846 inoculum; (8) *C. rosea* AN291 inoculum + *F. culmorum* KF846 inoculum.

#### 2.3.3. Inoculum Preparation and Inoculation

Inoculum of the pathogen and potential antagonists was produced by growing them on a PDA medium. Petri dishes with sporulating cultures were collected by washing with 10 mL sterile distilled water supplemented with 0.01% Tween 20 (Sigma-Aldrich, Steinheim, Germany). Spores were scraped with a sterile loop and shaken vigorously for several seconds. The concentration of the spores was determined by microscopic counting with a cell BRAND^®^ counting chamber (BLAUBRAND^®^ Bürker-Türk, Bristol, UK) and was adjusted to 1 × 10^5^ spores mL^−^^1^ in sterile water for *F. culmorum* KF846 and *Trichoderma*/ *Clonostachys* strains. All treatments were sprayed manually at 50 mL per row/500 mL per plot. In the semi-field experiment under covers, the spore suspension of antagonist strains was sprayed onto wheat spikes at the beginning of flowering stage (BBCH 63). The inoculation of the pathogen was performed at the full flowering stage (BBCH 65). In a field experiment, the spore suspension of *T. atrovirde* AN35 and *T. cremeum* AN392 was sprayed directly onto the kernels during sowing.

#### 2.3.4. Sampling and Selected Yielding Parameters Calculation

In semi-field three randomly selected wheat spikes were collected seven days after inoculation from each positive control plot-treated with a single fungus strain (*F. culmorum* KF846, *T. atroviride* AN240, *T. viride* AN430, *C. rosea* AN291) and subjected to microscopic analysis. At maturity 100 randomly selected wheat plants from each plot (field and semi-field trials) were collected: the spikes were threshed manually, thousand-kernel-weight and shoot biomass were measured and harvest index (HI) was calculated. The wheat kernels obtained from 100 spikes of the plot were bulked, weighed and subjected to mycotoxin analysis (field and semi-field trials) and to quantification of *F. culmorum* biomass (semi-field trials).

### 2.4. Mycotoxin Identification

#### 2.4.1. Chemicals and Reagents

The mycotoxins ZEN, DON, 3-AcDON, NIV and eniatins (Sigma-Aldrich, Steinheim, Germany) were dissolved in methanol (HPLC grade, Sigma-Aldrich, Steinheim, Germany) and used as standard stock solution. Water of HPLC grade from Merck Millipore water purification system was used in the analysis. Other chemicals used were all of analytical grade.

#### 2.4.2. Sample Preparation, Mycotoxins Extraction and HPLC Analysis

The samples from solid substrate cultures, the kernel samples from field and semi-field trials as well as the chaff samples from field trials were ground to a fine powder in the grinder. A total of 1 g of each powdered sample from the solid substrate cultures and the semi-field experiment was intended for quantitative analysis of *F. culmorum* biomass. Mycotoxins were extracted with acetonitrile:water (90:10, *v*/*v* or for enniatins 70:30, *v*/*v*) solution using a solvent mixture in a ratio of 2.5 mL of solvent per 1 g of the ground samples. Extractions were performed as described by Błaszczyk et al. [36]. Supernatants (1 mL) from liquid cultures supplemented with mycotoxin were diluted with HPLC grade methanol in 1:1 ratio of and filtered through a syringe filter (CHROMAFIL PET-45/15MS). The collected filtrates were analyzed by HPLC using the chromatographic system consisting of Waters 2695 high-performance liquid chromatograph (Waters, Milford, USA), Waters 2475 Multi λ Fluorescence Detector and Waters 2996 Photodiode Array Detector. Data were processed using Millenium software. DON and NIV analyses were performed according to the method described by Kostecki et al. [40] and Tomczak et al. [41] while 3-AcDON was detected by the method of Yang et al. [42]. Analysis of ZEN was performed according to Visconti and Pascale [43]. The limit of DON, NIV, 3-AcDON, ZEN, EnnB and EnnB1 detection was 0.01 μg g^−^^1^, 0.01 μg g^−^^1^, 3 μg kg^−^^1^, 1.0 g kg^−^^1^, 0.4 and 0.7 μg kg^−^^1^, respectively. Qualitative analysis of mycotoxins was carried out by comparing retention times and UV spectra of purified extracted samples with pure standards. Quantitative analysis of mycotoxins was performed by comparing peak areas of the analyzed samples to the calibration curve of peak areas obtained with mycotoxin standards.

### 2.5. QPCR Quantification of F. culmorum Biomass

Quantification of *F. culmorum* biomass was performed in samples from solid substrate (rice) cultures and in the kernel samples from semi-field trials by using a Real-Time PCR (qPCR) approach.

For the qPCR assay, DNA was isolated both from 50 mg powder-ground test samples and 50 mg *F. culmorum* powder mycelia as well as from 50 mg each serial dilution of mixture of *F. culmorum* powder mycelia and *F. culmorum*-free rice/wheat kernels, using the Wizard^®^ Genomic DNA Purification Kit (Promega, Madison, WI, USA). Homogeneous, powdered *F. culmorum* mycelium was obtained by culturing the KF846 strain on a PDA medium supplemented with 100 μg/mL streptomycin, lyophilization and grinding. Mixed samples were obtained from 10 mg of *F. culmorum* mycelia powder and 990 mg of *F. culmorum*-free rice/wheat kernel powder. A series of 10-fold dilutions (from 1 mg/g to 0.01 mg/g) of this mixture were prepared by sequentially combining 100 mg of each mixture with 900 mg of *F. culmorum*-free rice-wheat kernel powder. DNA obtained from pure mycelium and mixed samples was used initially to confirm the specificity and sensitivity of the qPCR assay and to optimize the reaction, including primer concentration and annealing temperature by analyzing the amplification plots, dissociation curves and excluding the primer dimers formation.

Finally, a Real-Time PCR was performed in 10 μL using 7.5 μL AmpliQ Real-Time PCR Opti Probe Kit (Novazym, Poland) and 100 nM of FAM-labelled probe and 300 nM of forward and reverse primers proposed by Waalwijk et al. [44]. Reactions were run using the C1000 Thermal Cycler with CFX96 Real-Time System (Bio-Rad Laboratories, Inc., Hercules, CA, USA). The established thermal cycling parameters were 2 min at 50 °C, 95 °C for 2 min followed by 40 cycles of 95 °C for 15 s and 60 °C for 30 s. Nuclease-free water (Merck KGaA, Darmstadt, Germany) was used as the no-template control. The positive control was *F. culmorum* genomic DNA. A standard curve was generated by plotting the Ct value for each sample of a standard series of the amount of fungal biomass. All samples were tested in triplicate.

### 2.6. Microscopic Analysis

#### 2.6.1. Pathogen-Antagonist Contact Zone Observation

Dual cultures of *F. culmorum* KF846 with *Trichoderma* or *Clonostachys* strains were used for microscopic observations. They were grown on a sterile strip of 20 mm cellophane membrane (50-μm thick) placed before fungal inoculation on a PDA medium in the middle of 8.5 cm Petri dish. Fungi were inoculated on opposite sides of Petri dishes at a distance of 5 mm from the edge. After 7 and 14 days of incubation at 25 ± 2 °C through 12 h/12 h of darkness/light, the mycelium overgrown cellophane membrane was cut with a razor blade in sterile conditions, placed on a microscope slide in a drop of distilled water and examined. Observations were carried out using a light microscope (Olympus CX-41-1 with UC-30 camera, Olympus Corporation, Tokyo, Japan). The samples were screened for loops of the *Trichoderma* or *Clonostachys* around *F. culmorum* hyphae and the anatomical damage of the pathogen.

#### 2.6.2. Fungi Wheat Kernel Colonization Observations

To estimate the ability of the *F. culmorum* KF846 and *T. atroviride* AN240, *T. viride* AN430, and *C. rosea* AN291 strains to colonize wheat kernels, scanning electron microscopic (SEM) observations were performed. In the semi-field experiment, two ears collected from each positive control plot and each plot inoculated with a single fungus strain were used for the analysis. Plant material for SEM was prepared exactly as described by Basińska-Barczak et al. [39] by fixation in a mixture of 4% methanol-free formaldehyde and 0.5% glutaraldehyde (Polysciences, Hirschberg an der Bergstrasse, Germany) with the addition of phosphate buffered saline (PBS). SEM imaging was performed using a Quanta FEG 250 microscope (FEI) in low vacuum mode at 70 Pa pressure and 10 kV beam accelerating voltage with 30 µm aperture, with a working distance of 10 mm.

### 2.7. Statistical Analysis

Statistical analyses of the experiments were performed using GenStat 19 Edition (the estimation of inhibition/reductions) and R software (other analyses and visualizations). The variation of measured variables was analyzed using linear mixed models and linear regression. In the models all effects and their interactions were assumed fixed, except for the replicates. Pearson correlation coefficients were calculated for analyses of time-dependent variables. For the growth inhibition heatmap, a hierarchical clustering with an average linkage method was used. The details of the statistical calculations are presented in the Appendix A.

## 3. Results

### 3.1. Assessment of the Antagonistic Potential of Trichoderma and Clonostachys Strains in a PDA Dual Culture Bioassay

Six *C. rosea* (AN272, AN291, AN295, AN297, AN298, AN300), five *T. atroviride* (AN152, AN182, AN206, AN240, AN497), four *T. viride* (AN255, AN401, AN430, AN826), three *T. harzianum* (AN150, AN279, AN360), two *T. citrinoviride* (AN262, AN393), *T. hamatum* (AN120, AN277), *T. koningiopsis* (AN143, AN251), *T. longibrachiatum* (AN197, AN213), *T. viridescens* (AN323, AN405) and single *T. cremeum* (AN392) and *T. longipile* (AN359) strains were screened for their antagonistic ability towards *F. culmorum* KF846 in dual cultures on PDA medium. The growth of *F. culmorum* KF846 was measured daily for 7 days of incubation in a single culture (control) and in co-incubation in dual culture with *Clonostachys* or *Trichoderma*. The mycelial growth of the *C. rosea* strains and *Trichoderma* species in the presence of *F. culmorum* was also measured daily for 7 days. The growth kinetics and the relationship in growth between the tested fungal strains are presented in Figure 1.

In order to determine the effect of *Trichoderma* and *Clonostachys* on the *F. culmorum* KF846 growth during 7 days of co-incubation, the % inhibition was estimated. The results of the *F. culmorum* KF846 growth reduction by *Trichoderma* and *C. rosea* strains are shown in Appendix A and illustrated in Figure 2. Slow-growing *Clonostachys* strains only slightly reduced the growth of Fusarium mycelium, namely between 10% and 46.67% on the first day of co-culture and between 15.29% and 23.53% on the last, seventh day of co-incubation. Meanwhile, at 2, 3 and 4 dpi, *C. rosea* growth was inhibited by the KF846 strain. Among the *Trichoderma* species, which, unlike *Clonostachys*, belong to the fungi with a fast growth rate, the most effective (70.59%) in inhibiting the *Fusarium* were: *T. atroviride* AN240, *T. koningiopsis* AN251 and *T. viride* AN430. Interestingly, after the first day of co-incubation, several *Trichoderma* strains, namely *T. atroviride* AN206, AN240 and AN497, *T. cremeum* AN392, *T. hamatum* AN277, *T. harzianum* AN159 and AN360, *T. longibrachiatum* AN197 and AN213, *T. longipile* AN359, *T. viride* AN401 and AN826, *T. viridescens* AN323 and AN504, showed a high percentage of inhibition of KF846 growth, ranging from 60 to 100%. As shown in the heat map (Figure 2), after 2, 3 and 4 days of co-culturing in a Petri dish with PDA, the impact of antagonistic strains on *F. culmorum* KF846 was lower than on the first day. However, in the next 3 days of co-incubation, the effectiveness of *Clonostachys* and *Trichoderma* strains increased. Finally, after 7 days of co-cultivation on a PDA medium, all tested *C. rosea* and *Trichoderma* strains showed the ability to limit the growth of *F. culmorum* ranging from 15.29 to 70.59%.

The qualitative evaluation of the ability of the *Trichoderma* and *Clonostachys* strains to antagonize *F. culmorum* KF846 was performed after 14 days of incubation in a dual culture. The modified scale of Mańka et al. [38] with a range from −8 (*F. culmorum* KF846 occupies more than 90% medium surface) to +8 (the antagonist occupies more than 90% medium surface) was used. The lowest score equal to −8 was recorded for the strains of *C. rosea* AN272, AN297 and AN300 (Figure 3, Appendix A). In turn, the highest score of +8 was obtained for Trichoderma strains AN240, AN251 and AN430, i.e., those with the greatest ability to inhibit the growth of *F. culmorum* KF846 (Figure 3).

These strains showed the ability to occupy an entire Petri dish with a PDA medium and to grow and sporulate on the pathogen mycelium (Figure 4). Regardless of this assessment, the contact zone between *F. culmorum* and the *Trichoderma* or *Clonostachys* strains grown in a dual culture on a PDA medium was observed microscopically. It was documented that the strains AN430 (Figure 4A) and AN35 (Figure 4B) as well as AN291 (Figure 4C) showed the mycoparasitic ability. As shown in Figure 4, after 7 (middle panel) and 14 (right panel) days of co-incubation, the antagonist hyphae coiled on *Fusarium* hyphae. After 14 days, the antagonist hypha was visible inside the partially degraded pathogen hypha (right panel A and C).

### 3.2. Assessment of the Potential of Selected Trichoderma and Clonostachys Strains to Reduce the Synthesis of Fusarium Mycotoxins in a Solid Substrate Bioassay

*Trichoderma* (AN240, AN430, AN251) and *Clonostachys* (AN291, AN298) strains with the highest potential to limit the growth of *F. culmorum* KF846 on PDA medium were selected for the dual culture bioassay on a sterilized rice kernel medium. After 21 days of co/incubation, the content of mycotoxins was determined in the post-culture (single strains)/dual culture/non-culture (non-inoculated rice medium) samples. The ability of *F. culmorum* KF846 to produce DON, 3-AcDON and ZEN was confirmed in the solid substrate bioassay. The presence of these mycotoxins was not found in single cultures of *Trichoderma* and *Clonostachys* strains. The results of the mycotoxin analysis in single *F. culmorum* KF846 cultures and dual cultures were used to estimate the percentage of mycotoxin reduction by antagonistic strains, calculated as a change of toxin content relative to the control. *Clonostachys rosea* AN291 and *T. atroviride* AN240 strains were observed as those with the highest potential (from 96.89 to 100%) to limit the accumulation of all tested mycotoxins on the rice medium (Appendix A, Figure 5). The *Trichoderma viride* AN430 strain was found to be capable of inhibiting the synthesis of DON and its 3-AcDON derivative in the range of 93.44 to 99.13% and ZEN by 94.77%, while *C. rosea* strain AN298 showed the ability to reduce DON, 3-AcDON and ZEN synthesis by 91.36%, 97.25% and 95.14%, respectively. *Trichoderma koningiopsis* AN251 has been documented as the weakest reducer of the mycotoxin content produced by KF846.

### 3.3. Assessment of the Ability of Trichoderma and Clonostachys Strains to Remove Mycotoxins in a Liquid Substrate Bioassay

Two *Trichoderma* strains (*T. atroviride* AN240 and *T. viride* AN430) and one *C. rosea* strain AN291 were selected for testing their ability to reduce the content of DON, 3-AcDON or ZEN toxins added to 48-h fungal cultures in a Czapek-Dox liquid broth. The changes in the content of ZEN, DON and its 3-AcDON derivative during the 96-h incubation of AN240, AN291 and AN430 in liquid culture are shown in Figure 6. In the culture of *C. rosea* AN291, a reduction in the amount of both ZEN and DON and 3-AcDON was observed after 1 h of incubation. *Clonostahys rosea* AN291 after 4 h of incubation with the addition of ZEN reduced the content of the toxin to the level of 0 (Figure 6, Appendix A). This strain after 96 h of incubation with the addition of 3-AcDON reduced its content to 0.04 ng/µL and with the addition of DON to 3.39 ng/µL (Figure 6, Appendix A). In the presence of *Trichoderma* strains, a decrease in the ZEN content in the medium was observed to the level of 0.32 ng/µL (AN430), 0.45 ng/µL (AN240) after 1 hpi and 0.02 ng/µL (AN240), 0.1 ng/µL (AN430) after 96 hpi (Figure 6, Appendix A). In contrast, strains AN240 and AN430 reduced trichothecenes type B to the amount of 0.41 ng/µL (3-AcDON), 7.03 ng/µL (DON) and 0.42 ng/µL (3-AcDON), 7.09 ng/µL (DON), respectively (Figure 6, Appendix A). The estimated linear regression coefficients indicate a significant trend in the decrease in the level of the analyzed toxins during (log time) incubation with each of the tested antagonistic strains in the liquid medium (Appendix A).

### 3.4. Toxin Transformations

In order to demonstrate the possibility of metabolizing DON to 3-AcDON and 3-AcDON to DON by the *C. rosea* AN291 strain, the content of both toxins was monitored during 96 h of incubation in a liquid medium supplemented with 0.5 ng/µL DON or 0.5 ng/µL 3-AcDON, respectively. A detailed comparison of the mutual content of DON and 3-AcDON in the culture medium with the DON supplement revealed cyclic changes in the concentration of these toxins from 0 to 96 hpi (Appendix A, Figure 7). Moreover, a significant strong positive correlation (r = 0.88, *p*-value = 4 × 10^−6^, ***) was observed between changes in DON -> 3ACDON concentrations in the AN291 culture with DON supplementation. A similar tendency to changes in the concentrations of both toxins was observed during the 8 h of incubation of AN291 with the addition of 3-AcDON (Appendix A, Figure 7). However, after 24 and 48 hpi, the content of 3-AcDON (0.4 ng/µL, 0.39 ng/µL) and DON (0.49 ng/µL) increased, and after 96 hpi reduced to the level of 0.27, 0.26. There was a weak negative correlation (r = −0.4, *p*-value = 0.06) between changes in 3AcDON -> DON concentration in AN291 culture with 3AcDON supplementation.

### 3.5. Semi-Field Experiment

*Trichoderma atroviride* AN240, *T. viride* AN430 and *C. rosea* AN291 strains showed an ability to inhibit the growth of *F. culmorum* KF846 on PDA and inhibit the synthesis of ZEN, DON and 3-AcDON by KF846 in solid cultures. Moreover, they were able to reduce the concentrations of endogenous mycotoxins in liquid cultures; therefore, these strains were used for the semi-field experiment.

Colonization of wheat kernels by these strains was initially estimated on the basis of microscopic observations (Figure 8). Scanning electron microscopy inspections showed an abundant growth of all tested strains on the surface of the kernel bran and kernel brush (Figure 8C–N1). The hyphae of *T. atroviride* AN240, *T. viride* AN430 and *C. rosea* AN291 were shown to “coil” on the hair of the kernel brush (Figure 8D,D1,G,G1,J,J1). The studied strains also sporulated on wheat kernels: *C. rosea* AN291 is visible on Figure 8I1, and *F. culmorum* KF846 on Figure 8M1. In addition, both the antagonist strains and *F. culmorum* KF846 showed the possibility of growing under bran (Figure 8E–E2,H,H1,K,K1,N,N1) on tube and cross cells. Nevertheless, no fungal hyphae were observed under the aleurone layer.

The wheat kernels obtained from 100 spikes of each plot (control and pre-/inoculated with *Trichoderma* or *Clonostachys* and *F. culmorum* KF846 strains, Bombona and Legenda cultivars, two seasons) were bulked, weighed and subjected to mycotoxin analysis. Changes in the weight of kernels obtained from 100 spikes for each experimental system, expressed as the percentage of change in relation to the control plants treated with water or plants inoculated with the pathogen alone, are presented in Appendix A and in Figure 9. It was observed that inoculation of ears with *Trichoderma* and *Clonostachys* strains prior to infection with *F. culmorum* KF846 resulted in an increase in weight of wheat kernel harvested from 100 ears for each variety and season ranging from 7.52% (Legenda, season I, AN240 pre-inoculation) to 63.65% (Legenda, season II, pre-inoculation AN430). Only in the Legenda cultivar, in season I, did pre-inoculation with strains AN291 and AN430 not increase the total mass of kernels harvested from 100 spikes.

Appendix A and Figure 10 show the changes in the toxin content in the kernels for each experimental system, expressed as a percentage change in relation to the control plants inoculated with the pathogen alone. It was shown that all strains are capable of reducing the content of mycotoxins produced by *F. culmorum* KF846, although its extent depended on the toxin, cultivar and growing season. Total inhibition (100%) of 3-AcDON synthesis was reordered for the Legenda cultivar in the first season after AN430 strain pre-inoculation and in the second season after AN291 strain pre-treatment. Meanwhile, the highest percentage (73%) of DON synthesis inhibition was determined for the Legenda cultivar in the first season after treatment with *T. atroviride* AN240. The highest percentage of ZEN reduction (47%) was recorded in the Bombona grain in the first season after AN240 treatment.

### 3.6. F. culmorum KF 846 Biomass in Solid (Rice) Medium and in Wheat Kernels

*Trichoderma atroviride* AN240, *T. viride* AN430 and *C. rosea* AN291 were assessed for their ability to reduce *F. culmorum* KF846 biomass under two experimental conditions: in samples from sterile rice cultivation and in kernels harvested from the semi-field experiment. In the antagonist-pathogen dual cultures on rice, a significant decrease in the biomass of *F. culmorum* KF846 was observed in relation to the amount of biomass in its single culture (Appendix A, Figure 11). Similarly, in a semi-field experiment, pre-inoculation of Bombona and Legenda wheat ears with each of the antagonist strains used contributed to a significant inhibition of *F. culmorum* KF846 growth in kernels. *Clonostachys rosea* AN291 was observed to be the most limiting strain of pathogen biomass growth in each experimental system.

### 3.7. Field Experiment

The field experiment was conducted with the use of two *Trichoderma* strains: *T. atroviride* AN35 and *T. cremeum* AN392. It was shown that the inoculation of wheat roots with the *T. cremeum* strain significantly increased the TKW parameter (Appendix A, Figure 12). In contrast, both *Trichoderma* strains significantly reduced HI. The content of DON, 3AcDON, NIV and ZEN in the kernels and chaff of control plants untreated with *Trichoderma* and plants treated with *Trichoderma* was also examined (Appendix A, Figure 13). In addition, the content of eniathins in the tested samples was analyzed (Appendix A, Figure 13). DON and 3-Ac DON were not found in kernel and chaff samples of control plants and *Trichoderma*-treated plants. Meanwhile, the presence of ZEN was found in control samples of cv. Bombona and those treated with *T. atroviride* AN35. In addition, the presence of ZEN was identified in the chaffs of Bombona and Legenda treated with AN35 and AN392. NIV was detected in most of the samples. Only in the kernels of the Legenda cultivar treated with AN392 and the chaffs of the untreated Legenda wheat plants was this toxin not identified. The presence of EnB and EnB1 was found in the kernels of both treated and control wheat cultivars. These metabolites were not found in the kernel chaff.

## 4. Discussion

### 4.1. PDA Dual Culture Bioassay: Preliminary Selection of Trichoderma and Clonostachys Strains Antagonistic to F. culmorum KF846

Initial assessment of antagonistic interactions between microorganisms involves the in vitro dual culture bioassay, often using a nutrient-rich PDA medium [45,46,47,48]. Recently, a bioassay with a dual PDA culture enabled the selection of several *Trichoderma* strains capable of complete overgrowth and sporulation on the mycelium of several *Fusarium* species [36,49,50]. In the current study, six *C. rosea* strains and 25 strains representing 10 *Trichoderma* species, namely *T. atroviride*, *T. viride*, *T. harzianum*, *T. citrinoviride*, *T. hamatum*, *T. koningiopsis*, *T. longibrachiatum*, *T. viridescens*, *T. cremeum* and *T. longipile*, were screened for their antagonism against the highly infectious and toxigenic *F. culmorum* strain KF846. These quantitative and qualitative analyses of the *Trichoderma*/or *Clonostachys*–*Fusarium* interaction allowed for the selection of three *Trichoderma* strains (*T. atroviride* AN240, *T. viride* AN430 and *T. koningiopsis* AN251) and two *C. rosea* strains (AN291, AN298) with the highest potential to limit the growth of *F. culmorum* KF846 on the PDA medium. Microscopic observations in the contact zone (Figure 4) showed the ability of strains AN240, AN430 and AN291 to surround pathogen hyphae and penetrate them. This effect has previously been recognized in *C. rosea* [26,51,52,53] and *Trichoderma* spp. [54,55,56,57] as a manifestation of mycoparasitism. Despite the different observed hyphal interactions with the pathogen, all strains that effectively inhibited the *F. culmorum* KF846 growth, namely *T. atroviride* AN240, *T. viride* AN430, *T. koningiopsis* AN251, *C. rosea* AN291 and *C. rosea* AN298, were selected for further study in the dual culture on solid rice medium.

### 4.2. Solid Substrate Dual Culture Bioassay: Selection of Trichoderma and Clonostachys Strains with the Ability to Reduce Mycotoxin Synthesis by F. culmorum KF846

High infectivity and pathogenicity to wheat as well as high toxigenic potential of the *F. culmorum* KF846 strain, originally isolated from wheat ears with symptoms of FHB, have been repeatedly documented [33,34]. This study confirmed that *F. culmorum* KF846 cultured on solid rice medium synthesized DON, 3AcDON and ZEN at concentrations of 33 µg/g, 4.6 µg/g and 41 µg/g, respectively. The presence of antagonistic fungi in the culture reduced the toxin content by 88.07 to 100%, depending on the *Trichoderma* species and the *C. rosea* strain. *Clonostachys rosea* AN291 and *T. atroviride* AN240 were considered the most effective in inhibiting the synthesis of DON, 3-AcDON and ZEN mycotoxins. The second reducer of *F. culmorum* KF846 toxins on the rice medium was *T. viride* AN430. Previously, both *T. atroviride* AN240 and *T. viride* AN430 showed the highest reduction potential for DON, 3AcDON and ZEN, but when co-cultured with another strain of *F. culmorum*, namely KF2795 [36]. However, data on the reduction in DON, 3AcDON and ZEN synthesis by *F. culmorum* as a result of co-incubation with *C. rosea* on sterile rice grains have not yet been disclosed. To our knowledge, the only reports evaluating the antagonistic potential of the *C. rosea* strain ACM941 (American Type Culture Collection ATCC 74447), isolated from pea plants in Manitoba, Canada in 1994, against *F. graminearum* in a dual culture bioassay were submitted by Hue et al. [58] and by Demissie et al. [59]. Meanwhile, Abdallah et al. [60] reported the activity of an endophytic strain of the genus *Clonostachys* in significantly inhibiting the growth of *F. graminearum* in PDA plate assays. Quantitative qPCR analyses showed that *T. atroviride* AN240, *T. viride* AN430 and *C. rosea* AN291 strains in rice dual cultures reduced the biomass of *F. culmorum* KF846 by approximately 52%, 42% and 75%, respectively. It can therefore be concluded that the reduction in toxin synthesis on rice medium by *F. culmorum* KF846 is related to the inhibition of the pathogen’s growth by *Trichoderma* and *Clonostachys* strains. Similar results were obtained by Matarese et al. [61], confirming the ability of *T. gamsii* 6085 to antagonize *F. graminearum* and *F. culmorum*, where it reduced the production of DON by up to 99% and, at the same time, the biomass of *Fusarium* species on haulms and rice substrate. According to Matarese et al. [61] the effect of *T. gamsii* 6085 is a direct reduction in the pathogen biomass (and thus the amount of DON). Therefore, the effect of *T. atroviride* AN240, *T. viride* AN430 and *C. rosea* AN291 on *F. culmorum* KF846 observed here can be explained in a similar way.

### 4.3. Liquid Substrate Bioassay: Determination of the Ability of Trichoderma and Clonostachys Strains to Eliminate Mycotoxins

The selected *T. atroviride* AN240, *T. viride* AN430 and *C. rosea* AN291 strains were used in further experiments aimed at determining their potential to reduce the content of endogenous DON, 3-AcDON or ZEN in liquid culture. The estimated linear regression coefficients indicated a significant decreasing of the analyzed toxin levels during (log time) incubation with each of the tested antagonist strains in the liquid medium. The ability of *Trichoderma* spp. to metabolize DON to the glycosylated form and to detoxify ZEN through sulfation has so far been demonstrated by Tian et al. [62,63], although this effect was studied on a solid PDA medium. No other reports have been found on the contribution of *Trichoderma* to the elimination of *Fusarium* mycotoxins. However, earlier works from the 1970s and 1990s indicated the ability of *Trichoderma* spp. [64,65] to degrade aflatoxin B1 (AFB1) produced by the molds *Aspergillus flavus* and *Aspergillus parasiticus*. The potential to metabolize mycotoxins produced by *Fusarium* sp., and more specifically by *F. graminearum*, has been extensively studied in *C. rosea*. In this species, ZEN detoxification was first described in 1988 by el-Sharkawy and Abul-Hajj [66]. Discovery of the enzyme zearalenone hydrolase (ZHD10) in *C. rosea* by Takahashi-Ando et al. [67] revealed the underlying mechanism of detoxification. Moreover, the mechanism of ZEN detoxification by the ZHD101 enzyme has been documented to be involved in mycoparasitic interactions between *C. rosea* and *F. graminearum*. Hence, this mechanism has been recognized as an attribute of biological control [68]. In the paper presented here, *C. rosea* AN291 after 4 h of incubation with the addition of 0.5 ng/μL of ZEN totally reduced the toxin content in the medium. The mechanism underlying this process is currently under investigation but is likely to be similar to that described by Takahashi-Ando et al. [67]. Meanwhile, after the first hour of incubation with the addition of 3-AcDON, the *C. rosea* AN291 strain decreased its content by about 36%, and after 96 h by about 92%. On the other hand, a significant DON reduction by AN291 was noted only after 96 h of incubation and was estimated to be around 58%. In a similar experiment, when *C. rosea* ACM941 was exposed to various concentrations of pure 15-ADON, significant evidence was provided that this strain is able to glycosylate 15-ADON and transform it into 15-ADON-3-Glc [59]. However, no data in the literature on DON transformation by *C. rosea* were found. Therefore, a detailed comparison of the mutual content of DON and 3-AcDON was made in this study during 96 h of cultivation of *C. rosea* strain AN291 in a liquid. Cyclic changes in the concentrations of these toxins over time were observed, with a significantly strong positive correlation between changes in DON -> 3-AcDON concentrations during the 96-h AN291 culture with DON supplementation and a negative correlation between changes in 3-AcDON -> DON concentrations during the first 8 h of AN291 culture with 3-AcDON supplementation. However, in the following hours of AN291 culture with the addition of 3-AcDON, this correlation was positive and analogous to that observed in the culture of *C. rosea* AN291 with the addition of DON. The explanation of this occurrence, however, requires further detailed research and careful consideration, especially in terms of mycoparasitism against *F. culmorum* and aspects of biological control with AN291.

### 4.4. Model Semi-Controlled Field and Field Experiment: The Effect of Trichoderma and Clonostachys on the Accumulation of Fusarium Mycotoxins in Wheat Tissues

In the presented work two field experiments were carried out: semi-controlled field conditions and field conditions.

In the semi-controlled field experiment, the ability of *T. atroviride* AN240, *T. viride* AN430, *C. rosea* AN291 and *F. culmorum* KF846 strains to colonize wheat kernels was confirmed by scanning electron microscopy inspections. Interestingly, the antagonistic strains were able to form “coiled coil-like” structures on the hair of the kernel brush (Figure 8), described as a characteristic feature of mycoparasitic fungi [27,69]. The formation of dense coils around the hyphae of pathogen was observed both in this study in a dual culture experiment on agar medium, as well as in previous own [36] and other authors’ works [70,71,72]. It is noted that this feature considered to be characteristic of mycoparasitism may also manifest itself in symbiotic interactions with plants, as documented by Abdellatif et al. [46] who studied the interactions of endophytic *Ascomycota* fungi with wheat roots. The current microscopic observations showed that both the antagonist strains and the *F. culmorum* KF 846 strains grew under bran (Figure 8) on tubes and crosses, but their hyphae were no longer discernible under the aleurone layer. Despite numerous works on the kernel treatment of crop plants, to our knowledge, the degree and extent of colonization of kernels harvested from wheat ears treated with *Trichoderma* and *Clonostachys* by scanning electron microscopy has not been described so far. These studies were carried out on spring wheat kernels infected by *F. culmorum* [73]. Jackowiak et al. [72] noted the presence of *F. culmorum* hyphae in mature kernels of spring wheat mainly in the area of the crease, including the brush end, the outer layer of the kernel and in the area between the coat and the aleurone layer and in the aleurone layer itself.

Wheat kernels collected from all variants of the semi-controlled field experiment tested here were analyzed for the total weight of kernels harvested from 100 ears. In samples of these kernels, the content of the mycotoxins and the *F. culmorum* biomass were also estimated. It was observed that pre-inoculation with *Trichoderma* and *Clonostachys* strains inhibited the accumulation of all *F. culmorum* KF846 mycotoxins and stimulated the total kernel weight of both the spring cultivar Bombona and the winter cultivar Legenda in both growing seasons. The only exception were samples after pre-inoculation with strains AN291 and AN430 in the first season, where a decrease in total grain weight was observed in the Legenda cultivar. In order to demonstrate that the observed decrease in the content of mycotoxins in the wheat grains is accompanied by inhibition of the growth of *F. culmorum* KF846, an attempt was made to determine the biomass of *F. culmorum* using the qPCR. It was documented that the initial inoculation of Bombona and Legenda wheat ears with each of the applied antagonist strains significantly reduced the biomass of *F. culmorum* KF846. A similar regularity was noted here in the bioassay on a solid rice medium. It is worth emphasizing that both in vitro and in the semi-controlled field experiment, *C. rosea* AN291 was the most effective in limiting the growth of pathogen biomass. Previous studies have indicated that the effect of *Trichoderma* pre-inoculation is not stable over time and may depend on climatic conditions. Sarocco et al. [73] and Alukumbura et al. [74] described the variable effects of the pre-inoculation of *T. gamsii* T6085 wheat ears on the frequency and severity of FHB caused by *F. graminearum* and wheat agronomic parameters. Based on quantitative analysis, Alukumbura et al. [74] noted that pre-inoculation with *T. gamsii* T6085 did not limit the abundance of *F. graminearum* in wheat ears/kernels, although disease rates were reduced. Meanwhile, Gimeno et al. [51] analyzed the effect of *C. rosea* on reducing the severity of wheat disease caused by *F. graminearum*. In a controlled experiment, these authors showed a reduction in DON contamination by *C. rosea* in the range of 42 to 82%, and in the field experiment in the range of 45 to 69%. However, in the context of the content of ZEN in grains, after inoculation of *C. rosea*, they noted either an increase in these toxin-controlled conditions or no significant effect on its content field conditions. The role of the preparation and the concentration of spores as well as the time of their use were also discussed [58,75]. In a study by Hue et al. [58] *C. rosea* strain ACM941 applied to wheat against FHB caused by *F. graminearum* in greenhouse experiments reduced the accumulation of DON by 21%. The same ACM941 strain, but in the form of the biofungicide CLO-1 in greenhouse conditions reduced the DON content in kernel by 51–95%, but in field conditions by 22–33% [51], comparable to the results in earlier studies of Hua et al. [58] using an unformulated *T. gamsii* strain. Here it was noted that the extent of the pre-inoculation effect could depend on the *Trichoderma* and *Clonostachys* strain, wheat variety and growing season. Since the experiment was carried out in two growing seasons, but under covers, it is not assumed that the influence of weather conditions was as significant as in the field experiments. Meanwhile, little is known about the effect of pre-treatment of wheat ears with *Trichoderma* or *Clonostachys* on both *Fusarium* biomass (abundance) and mycotoxin content. This knowledge is rather limited to the data from in vitro experiments already cited above. Therefore, the results obtained in this work seem to be new and important for further considerations on the use of *Trichoderma* or *Clonostachys* strains in a biological control consisting of limiting the growth of *F. culmorum* and the accumulation of its mycotoxins in wheat kernels.

The second model experiment was conducted in field conditions with the use of two *Trichoderma* species: *T. atroviride* AN35 isolated from grains endosphere [28] and *T. cremeum* AN392 isolated from decaying wood [29]. Both of these strains proved to be effective producers of lytic enzymes and AN392 especially was considered the most effective producer of cellulolytic and xylanolytic enzymes [29]. In addition, these strains were previously analyzed for the production of volatile metabolites, and on this basis AN35 was considered the most effective producer of volatiles, including antimicrobial 6-npentyl-2H-pyran-2-one [76]. *Trichoderma atroviride* AN35 was also found to be the most effective mycoparasite of the mycotoxigenic species *F. culmorum*, *F. graminearum* and *F. avenaceum* [76,77,78]. Hence, these two strains with such a different lifestyle, from saprophytic in AN392 to strongly mycoparasitic in AN35, were introduced here for field research. These strains were introduced to the field experiment by spraying kernels at sowing. The effect of soil inoculation with AN35 and AN392 was assessed by analyzing the content of toxins most frequently detected in wheat tissue in Poland [79,80], namely DON, 3-AcDON, 15-AcDON, NIV, ZEN and eniathins. The obtained results indicated that soil inoculation during sowing of winter and spring wheat does not directly determine the reduction in *Fusarium* mycotoxin accumulation in ears. Interestingly, such a direct effect of soil treatment with *T. cremeum* AN392 was observed in the case of the TKW yield parameters of both cultivars, which may indicate a higher healthiness of the kernels of the treated plants compared to the control plants. In addition, both strains reduced the HI, which suggests that the *Trichoderma* soil treatment stimulates the amount of plant biomass and not necessarily its yield. The available literature describes many examples of plant growth improvement by the action of *Trichoderma* [27,81]. In particular, this effect has been noted in terms of root growth promotion, although a significant increase in above-ground plant parts such as stem length and thickness, leaf area, chlorophyll content as well as yield has also been observed [81]. There are reports documenting that *Trichoderma* treatment can also improve wheat yield by 6–11% [82,83]. Although the results presented here do not directly indicate an enhancement in wheat yield, the observed increase in the TKW parameter indicates an improvement in seed health. Meanwhile, a decrease in the HI parameter only suggests a higher ratio of plant biomass to yield. It can be assumed that an increase in biomass (although unfavorable from an economic point of view) indicates greater plant vigor and, consequently, plant health. Moreover, as evidenced by the previously cited literature reports [58,73,74], the effect of inoculation with *Trichoderma* fungi in field conditions could depend on weather conditions. According to locally monitored data (2017/2018 in Cerekwica, Western Poland; GPS coordinates: 52°5′ N, 16°7′ E), the temperature during the sowing of winter wheat Legenda in the second half of September 2017 was T_max_ 17 °C and T_min_ 13 °C, with precipitation of 3 mm. The temperatures recorded during spring wheat Bombona sowing at the end of March 2018 were T_max_ 7°C and T_min_ 0 °C, with precipitation of 7 mm. The average temperature at full flowering (BBCH 65) was, respectively, for the Legenda at the rainfall of 0 mm: T_max_ 24 °C and T_max_ 8 °C, and for the Bombona at the precipitation of 4 mm: T_max_ 25 °C and T_max_ 13 °C. Hence, with more favorable weather conditions during the flowering of the spring form, the cv. Bombona had a higher content of NIV and ZEN toxins, both in the ears and in the chaff, compared to the values obtained for the Legenda cultivar. However, in order to assess the correlations between the weather conditions during sowing and flowering, which are crucial for the development of fungi, and the impact of beneficial microorganisms, several years of field experiments should be carried out. Therefore, the fungal strains selected on the basis of the presented research are planned to be used to develop formulations containing these strains, which can be used in multi-area wheat crops. The planned research will therefore require several years of field observation and analysis of various factors determining the effectiveness of the bio-formulation being developed.

## 5. Conclusions

The quantitative and qualitative analyzes of the *Trichoderma*/or *Clonostachys*-*Fusarium* interaction allowed for the selection of three *Trichoderma* strains (*T. atroviride* AN240, *T. viride* AN430, *T. koningiopsis* AN251) and two *C. rosea* strains (AN291, AN298) with the highest potential to limit the growth of *F. culmorum* KF846 on PDA medium. *Clonostachys rosea* AN291, *T. atroviride* AN240 and *T. viride* AN430 were considered the most effective in inhibiting the synthesis of *F. culmorum* KF846 mycotoxins—DON, 3-AcDON and ZEN, when co-cultured on a solid rice medium. It has been documented that pre-inoculation of wheat ears with these strains also reduces the accumulation of *F. culmorum* mycotoxins in grains. Quantitative qPCR analysis showed that the presence of *T. atroviride* AN240, *T. viride* AN430 and *C. rosea* AN291 both in dual solid cultures and after pre-inoculation of wheat ears in semi-field experiments reduced the biomass of *F. culmorum* KF846. It can therefore be concluded that the reduction in toxin synthesis by *F. culmorum* KF846 is related to the inhibition of pathogen growth by *Trichoderma* and *Clonostachys* strains. The experiment on liquid culture with the addition of DON, 3-AcDON and ZEN showed that selected strains also have the potential to eliminate toxins from the culture medium, which may indicate the ability of these strains not only to reduce the production of mycotoxins by *Fusarium*, but also to metabolize them. However, it was found that soil inoculation with *Trichoderma* species during sowing of winter and spring wheat did not directly reduce the accumulation of *Fusarium* mycotoxins in ears. The direct effect of soil treatment of plants was observed in the case of two yielding parameters, the values of which indicated a higher healthiness of seeds and an increase in the biomass of plants treated with *Trichoderma* species. It can therefore be concluded that the selected *Trichoderma* and *Clonostachys* strains have a high potential to reduce the content of *F. culmorum* mycotoxins in wheat tissues and improve crop health.

## Figures and Tables

**Figure 1 jof-09-00289-f001:**
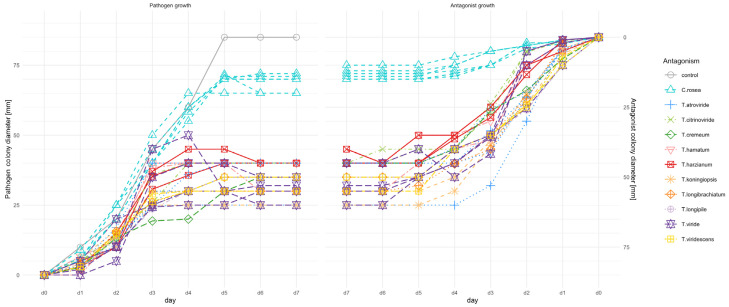
Visualization of reciprocal antagonist (*Trichoderma* species or *C. rosea* strains) vs. pathogen (*F. culmorum* KF846) growth on PDA medium during 7 days of incubation/co-incubation. Control-growth of *F. culmorum* in a single culture.

**Figure 2 jof-09-00289-f002:**
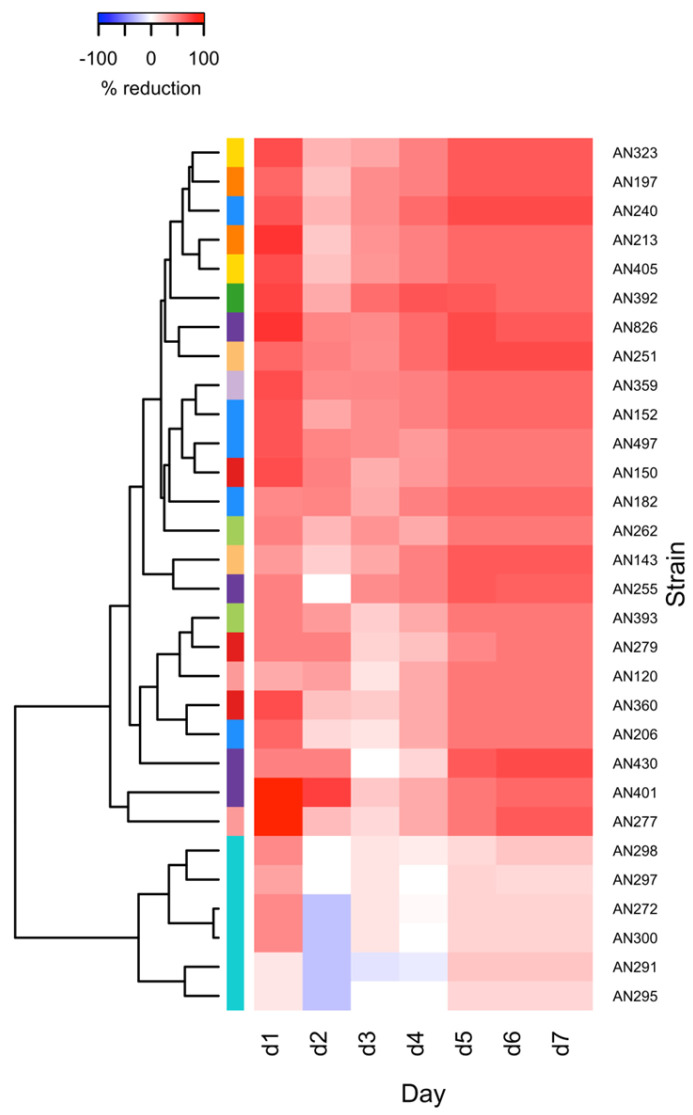
Heat map of *F. culmorum* KF846 mycelial growth inhibition (%) induced by *Trichoderma* or *Clonostachys* strains during 7 days of co–incubation in dual cultures on PDA medium.

**Figure 3 jof-09-00289-f003:**
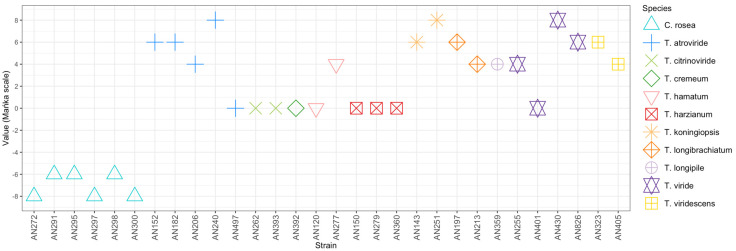
The qualitative evaluation of the *F. culmorum* KF846—*Clonostachys* and *F. culmorum* KF846—*Trichoderma* interaction after 14 days of co-incubation on PDA medium by using modified Mańka [36]. The symbol represents a median of three replications.

**Figure 4 jof-09-00289-f004:**
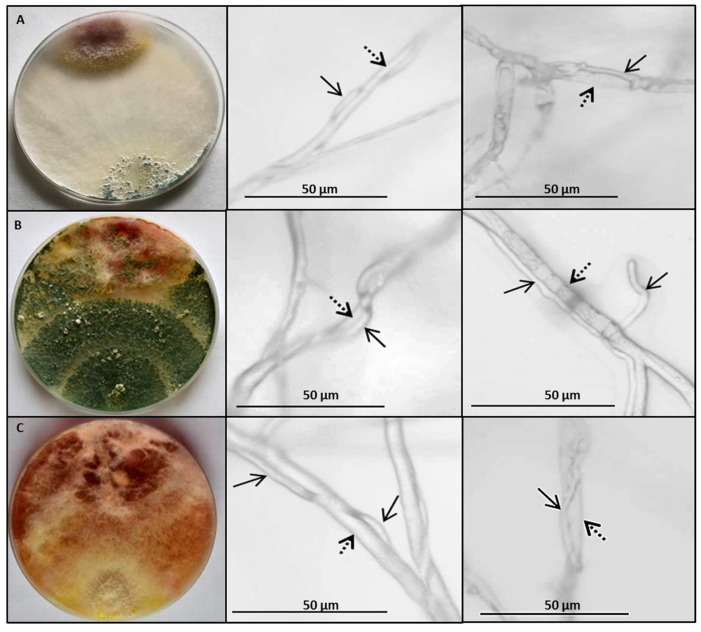
Interactions between *F. culmorum* KF846 and (**A**) *T. viride* AN430; (**B**) *T. atroviride* AN240; (**C**) *C. rosea* AN291 in dual cultures on a PDA medium. The left panel shows the growth of the pathogen and antagonist co-cultured in Petri dishes, the middle panel shows microscopic observations of the contact zones of pathogen and antagonist after 7 days of co-incubation and the right panel shows microscopic observations after 14 days of fungi co-incubation. The dashed arrow points to the hyphae of *F. culmorum* KF846 and the solid arrow points to the hyphae of *Trichoderma* or *Clonostachys*.

**Figure 5 jof-09-00289-f005:**
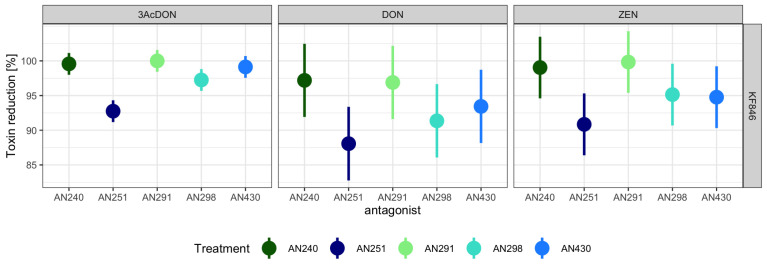
Toxin reduction (%) as a change of 3-AcDON, DON and ZEN content (pathogen-antagonists dual culture) relative to the control culture (*F. culmorum* KF846).

**Figure 6 jof-09-00289-f006:**
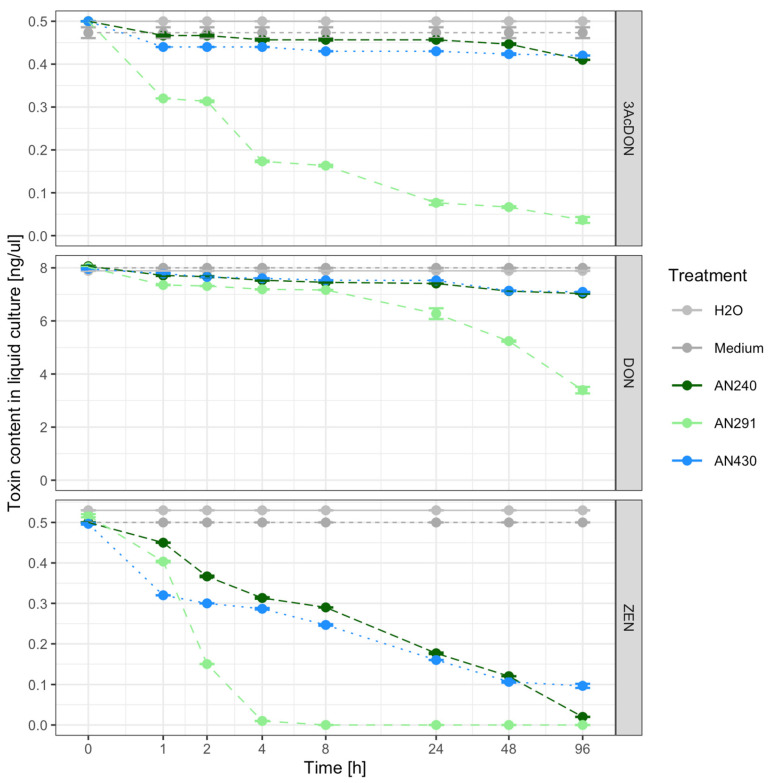
Timeline plot. Toxin content in liquid solution, depending on time (1, 2, 4, 8, 24, 48, 96 hpi) and treatment (H_2_O, medium, AN240, AN291, AN430).

**Figure 7 jof-09-00289-f007:**
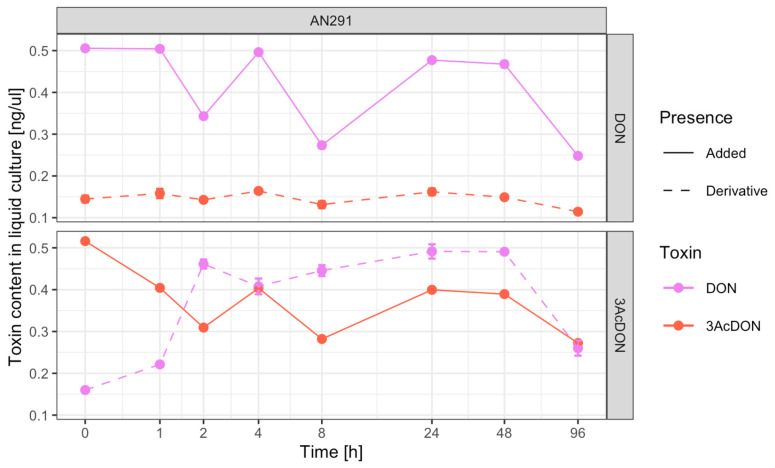
Changes in the content of DON and 3-AcDON mycotoxins in parallel during 96 h of culturing the *C. rosea* AN291 strain in a liquid medium with the addition of 3AcDON or DON.

**Figure 8 jof-09-00289-f008:**
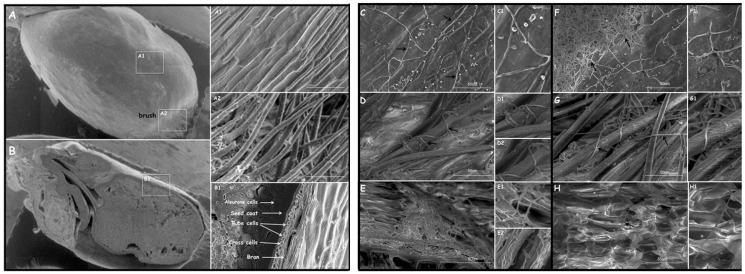
Electron scanning micrography of control wheat kernels not inoculated and treated with *Trichoderma* and *Clonostachys* and not infected with *F. culmorum* KF846. On panel (**A**,**B**) are control kernels of wheat. On panels (**C**–**E**) the wheat kernel treated with *T. viride* AN430 (black arrows) are visible. Panels (**F**–**H**) present the wheat kernel treated with *T. atroviride* AN240 (black arrows). On panels (**I**–**K**) the kernels treated with *C. rosea* AN291 (black arrows) are presented. Panels (**L**–**N**) show the kernel treated with *F. culmorum* KF846 (black arrows). In each panel bran, brush and section are presented, respectively: (**A1**,**C**,**C1**,**I**,**I1**,**L**,**L1**)—magnification on bran; (**A2**,**D**,**D1**,**D2**,**G**,**G1**,**J**,**J1**,**M**,**M1**,**M2**)—magnification of brush; (**B1**,**E**,**E1**,**E2**,**H**,**H1**,**K**,**K1**,**N**,**N1**)—magnification on wheat kernel section.

**Figure 9 jof-09-00289-f009:**
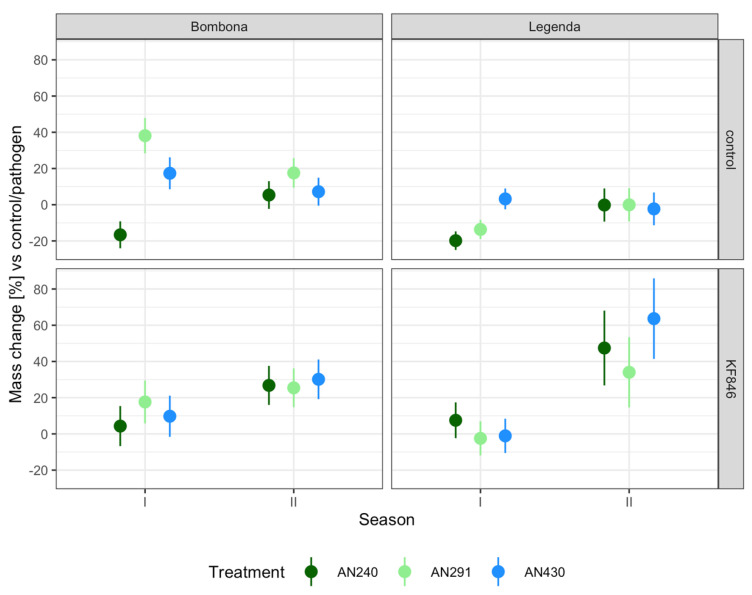
Yield mass changes in semi-field experiments. Mass % increase/decrease scores.

**Figure 10 jof-09-00289-f010:**
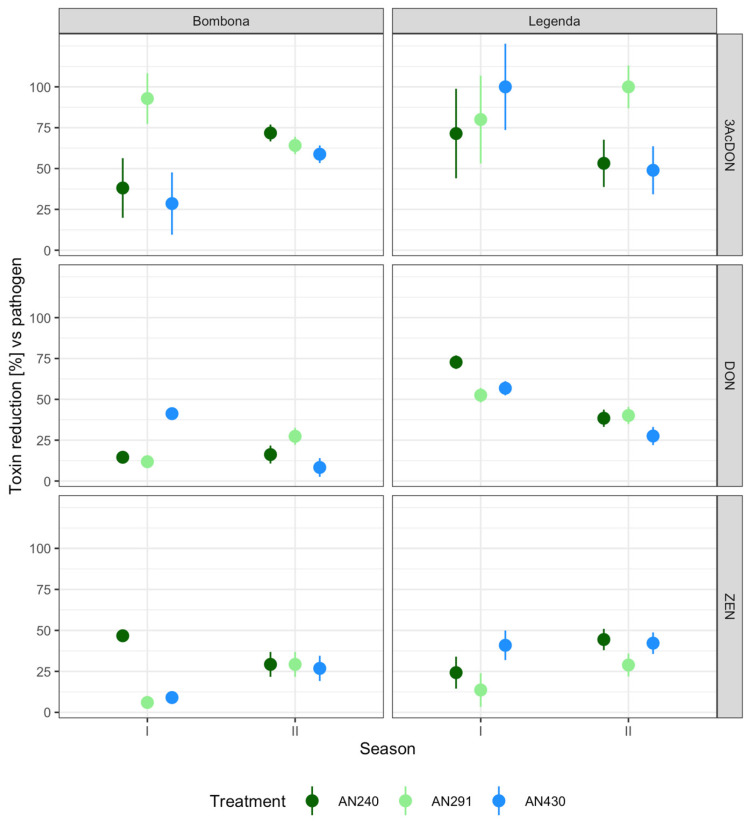
The differences (%) in toxin contents compared to the control (inoculation by pathogen only).

**Figure 11 jof-09-00289-f011:**
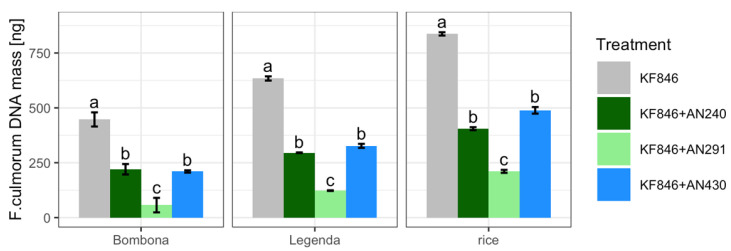
Biomass of *F. culmorum* determined by qPCR in culture/co-culture on rice solid medium and in wheat kernels (Bombona, Legenda) inoculated with the pathogen (KF846, control) and inoculated with both the pathogen (KF846) and the antagonist (AN240 or AN291 or AN430). Letters indicate significance groups within the row (for each yield parameter) according to Tukey’s test at 0.05.

**Figure 12 jof-09-00289-f012:**
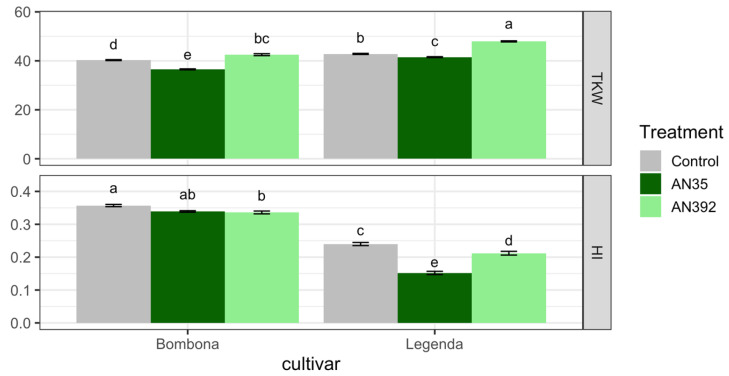
Comparison of TKW and HI in control wheat plants and after *Trichoderma* root inoculation. Letters indicate significance groups within the row (for each yield parameter) according to Tukey’s test at 0.05.

**Figure 13 jof-09-00289-f013:**
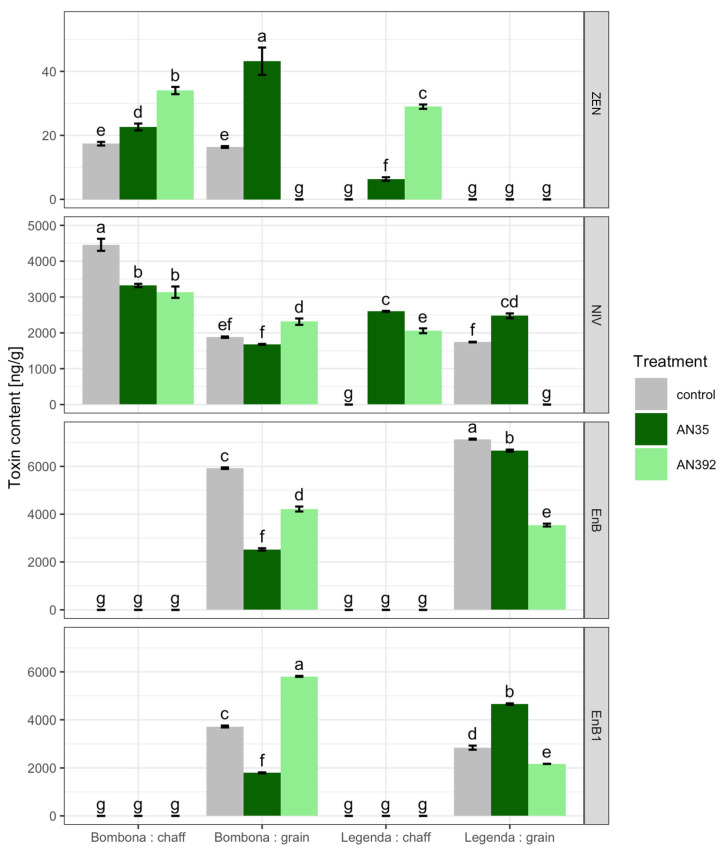
Toxin content in wheat kernels after *Trichoderma* root inoculation. Letters indicate significance groups within the row (for each toxin) according to Tukey’s test at 0.05.

## Data Availability

Data supporting the reported results are provided in Appendix A.

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
