# Peer review of "Containment of Fusarium culmorum and Its Mycotoxins in Various Biological Systems by Antagonistic Trichoderma and Clonostachys Strains"

_jof, 2023, doi:10.3390/jof9030289_

Round 1
Reviewer 1 Report
The present manuscript entitled « Containment of Fusarium culmorum and its mycotoxins in various biological systems by antagonistic Trichoderma and Clonostachys strains” investigates the biocontrol or biostimulation capacities of strains belonging to the Trichoderma and Clonostachys genera using both in vitro and in field approaches.
The experiments are well designed, using appropriate controls and relevant statistical analyses. Results are properly presented and most of the time well discussed, with no over-interpretation. The manuscript is overall well written.
This work therefore deserves publication in Journal of Fungi provided minor revisions (listed below) are brought to the manuscript.
Minor revisions:
Main text
In the introduction, the last paragraph comes abruptly. A short mention to the two concerned fungal genera and previous knowledge on their biological properties should be added.
In the results part, part 3.3 (lanes 400-422): the HPLC method used for the detection of mycotoxins is most likely not sufficient to detect conjugates or other mycotoxins derivatives which would require coupling with mass spectrometry. The unexplained results, notably the negative correlation observed when using 3-ADON as the original form may be explained by the lack of precision of the analytical method. The authors should discuss this point. In addition, no error bars are apparent on the graph. Does this mean that the experiment was only performed once? If so, the authors should be more cautious in their conclusions. Finally, the authors do not discuss potential chemical transformation of the mycotoxins (even indirectly due to the medium modification (e.g. pH) by the fungal strain(s)). This could be added in the discussion.
Concerning field experiments, having weather data would be helpful to better understand/interprete the results. Regarding the biostimulation assays, are there other phenotypic data available (spike number for example)? This would be a plus to conclude about the major role of fungal applications in the vegetative phase.
In the discussion, notably elements about field experiments (part 4.4 – lines 630-749) is difficult to follow and should be better structured and more concise.
Tables and figures
Figure 1: curves are difficult to follow due the high number of strains
Figure 8: images N and N1 are not fully clear to me (do they really correspond to sections?).
Table S4b: is this table correctly numbered? It refers to growth inhibition and not mycotoxin content.
Captions could be completed (colour codes, reference to seasons I et II in supplementary tables, …).
Typing/spelling mistakes:
In text and figures, homogenize ZEA/ZEN
Figure 1, y axis: colony --> colony.
Lane 451: coiling --> coil.
Lane 480: the term “reduction” is not appropriate.
Lane 705: Alukumbur --> Alukumbura.
Line 726: mytoxic -->à mycotoxigenic.
Author Response
Dear Reviewer 1,
We appreciate the time you take to read our article and thank you for your valuable comments that may have led to improvements in the current version. We tried to carefully consider all comments and we made every effort to respond to each of them.
Changes made are displayed with a yellow highlight in the Word version of the manuscript. Moreover, below we have answered point by point to the Reviewer's comments.
We hope the Reviewer will find our answers satisfactory and finding the manuscript acceptable for further publication procedures.
Kind regards,
Lidia Błaszczyk
Response to Reviewer 1 suggestions:
- In the introduction, the last paragraph comes abruptly. A short mention to the two concerned fungal genera and previous knowledge on their biological properties should be added.
Suggested data are listed in the introduction, before the last paragraph on the research objective.
- In the results part, part 3.3 (lanes 400-422): the HPLC method used for the detection of mycotoxins is most likely not sufficient to detect conjugates or other mycotoxins derivatives which would require coupling with mass spectrometry. The unexplained results, notably the negative correlation observed when using 3-ADON as the original form may be explained by the lack of precision of the analytical method. The authors should discuss this point. In addition, no error bars are apparent on the graph. Does this mean that the experiment was only performed once? If so, the authors should be more cautious in their conclusions. Finally, the authors do not discuss potential chemical transformation of the mycotoxins (even indirectly due to the medium modification (e.g. pH) by the fungal strain(s)). This could be added in the discussion.
Thank you very much for the above suggestions. We are aware that the data presented by us as a result of HPLC analysis cover only the aspect of identification/non-identification of the tested toxin in liquid culture. Hence the title of subsection 3.3. Nevertheless, this analysis showed that the tested strains "reacted" to the presence of the toxin. Towards understanding the cause of "removal" of the toxin from the environment - accumulation in mycelium, biotransformation, or even detoxification - we have already started research. We described in 4.3 subchapter that solving this problem will be the subject of our further research.
In the Discussion chapter, subchapter 4.3, we have therefore limited ourselves to the use of the phrases "toxin removal" or "toxin elimination" in order to avoid overinterpretation and inadequate conclusions, without knowing the exact mechanism of this phenomenon yet. We also referred to earlier work on the chemical transformation of mycotoxins by Clonostachys and Trichoderma species. Our intention, however, was to avoid a broad discussion (with a fairly large manuscript volume) on the topic of mycotoxin masking, possible pathways of their metabolism by Trichoderma and Clonostachys, etc., because, as the Reviewer mentioned, such analyzes have not been performed in the current work.
We would also like to clarify that these experiments were performed in triplicate biological replicates. Perhaps due to the type of graphs and the style of the error bars, they are not noticeable, but they are present in Figures 6 and 7. Detailed data and calculations are also presented in Appendix Supplementary_results file.
- Concerning field experiments, having weather data would be helpful to better understand/interprete the results. Regarding the biostimulation assays, are there other phenotypic data available (spike number for example)? This would be a plus to conclude about the major role of fungal applications in the vegetative phase.
Of course, we agree with the Reviewer that the analysis of weather conditions would support the interpretation of the results. Therefore, in the Discussion chapter, subchapter 4.4, we quoted the weather conditions at the key sowing dates (winter and spring wheat) and during flowering, when the crops are most exposed to Fusarium infections.
As the presented field experiment was a model and covered two seasons, we did not undertake a statistical analysis of the impact of weather conditions on the observed effect, i.e. the healthiness of kernels and the content of Fusarium toxins in them.
Answering the question regarding the observation of other phenotypic traits in wheat plants treated with Trichoderma, we would like to explain that such analyzes were of course carried out, both in the field and in greenhouse and sterile conditions - in vitro. The aim was to understand the molecular basis of wheat's response to Trichoderma spp. root colonization. We analyzed changes in the morphology, anatomy, physiology of plants, as well as the proteome, transcriptome and metabolome of wheat plants, caused by root colonization by two Trichoderma species, with particular emphasis on identifying unique and specific changes depending on the fungal strain. We noted significant changes in the tested parameters caused by the colonization of plants by Trichoderma. The results of these studies are being prepared for publication. In the current work, we focused on the subject only on the effect of Trichoderma inoculation on the health of kernels, and thus on the release of wheat grains from Fusarium colonization and accumulation of toxins.
- In the discussion, notably elements about field experiments (part 4.4 – lines 630-749) is difficult to follow and should be better structured and more concise.
We have rewritten this section, also adding information about the weather data mentioned above.
- Tables and figures
- Figure 1: curves are difficult to follow due the high number of strains
We agree with the Reviewer's opinion. This figure is only illustrative, while detailed data are included in Appendix Supplementary_results file.
- Figure 8: images N and N1 are not fully clear to me (do they really correspond to sections?).
Each panel: [C, D, E ], [F, G, H], [I, J, K], [L,M,N] shows bran, brushes and sections, respectively. Hence, in each panel, photographs E and E1, H and H1, K and K1 and N and N1 show sections of the kernel at the appropriate magnification. A detailed cross section of the control kernel is shown in photos B and B1.
- Table S4b: is this table correctly numbered? It refers to growth inhibition and not mycotoxin content.
Thank you for this attention. Of course, this chart should be numbered Table S2b.
- Captions could be completed (colour codes, reference to seasons I et II in supplementary tables).
In the supplementary tables (field trials), we have marked the boxes corresponding to the graphs in the manuscript to make it easier to track the data.
- Typing/spelling mistakes:
- In text and figures, homogenize ZEA/ZEN
The abbreviation ZEN has been introduced both in the tables and in the text.
- Figure 1, y axis: colony --> colony.
The error has been corrected.
- Lane 451: coiling --> coil.
The error has been corrected.
- Lane 480: the term “reduction” is not appropriate.
The term "reduction" has been rewritten as "changes"
- Lane 705: Alukumbur --> Alukumbura.
Name error has been corrected.
- Line 726: mytoxic -->à mycotoxigenic.
The error has been corrected

Reviewer 2 Report
Dear Authors,
The MS is well written and presented in an organised way. In my opinion, the MS can be accepted for publication. I have small query authors mentioned 33 isolates (1 + 6+27) which should be thirty four whereas in discussion the number is 6+25.
Author Response
Dear Reviewer 2,
We appreciate the time you take to read our article and thank you for your comment. We would like to explain that in the work we used 32 strains: 1 Fusarium strain., 6 Clonostachys strains. and 25 Trichoderma strains. We corrected these data accordingly in the text. Changes made are displayed with a yellow highlight in the Word version of the manuscript.
We also thank you for recognizing our work as acceptable for publication.
Kind regards,
Lidia Błaszczyk

Reviewer 3 Report
In this paper, Błaszczyk and co-workers present the selection of Trichoderma and Chlonostachys strains by their ability to reduce the growth and mycotoxin contamination of the FHB pathogen F. culmorum in different artificial and natural substrates, and by their ability to enhance the plant fitness in terms of grain weight and harvest index. They also examine the ability of some selected strains to metabolize F. culmorum mycotoxins. They find a number of potential strains that can be further used as biocontrol agents against F. culmorum. Overall, the work is complete, controls and replicates are well organized and experiments are well designed including field trials. I would like to note:
- -MS needs English grammar check and correction of typing errors either in the main text and in the references list.
- -Reference Nº 7 may be a little obsolete in 2023…at least if the authors are referring to the last decade.
- -Reference Nº 72 (Benítez et al., 2004) in the discussion section results a bit aged to be used as a reference paper for Trichoderma General Mechanisms of Action, since there is Woo et al (Nat. Rev. Microbiol., 2022).
- -In M&M section qPCR experiment design, authors specify how they prepare samples for standard curve construction. It would be helpful for readers if authors explain also test sample preparation from that 1mg of powder, the total gDNA concentration used for qPCR reactions, primers used and if an efficiency test/exclusion of dimmer formation has been performed in their work without needing to jump to reference 45!
Author Response
Dear Reviewer 3,
We appreciate the time you take to read our article and thank you for your valuable comments that may have led to improvements in the current version. We tried to carefully consider all comments and we made every effort to respond to each of them.
Changes made are displayed with a yellow highlight in the Word version of the manuscript. Moreover, below we have answered point by point to the Reviewer's comments.
We hope the Reviewer will find our answers satisfactory and finding the manuscript acceptable for further publication procedures.
Kind regards,
Lidia Błaszczyk
Response to Reviewer 3 suggestions:
- MS needs English grammar check and correction of typing errors either in the main text and in the references list.
We reviewed the manuscript in accordance with the guidelines. We hope that we have detected all irregularities.
- Reference Nº 7 may be a little obsolete in 2023…at least if the authors are referring to the last decade.
The problem of diseases caused by F. culmorum on wheat crops in New Zealand continued. Accordingly, the data was updated and we referred to a similar publication from 2006: “Bentley, A.R., Cromey, M.G., Farrokhi-Nejad, R. et al. Fusarium crown and root rot pathogens associated with wheat and grass stem bases on the South Island of New Zealand. Australasian Plant Pathology 2006, 35, 495–502, doi.org/10.1071/AP06053”.
- Reference Nº 72 (Benítez et al., 2004) in the discussion section results a bit aged to be used as a reference paper for Trichoderma General Mechanisms of Action, since there is Woo et al (Nat. Rev. Microbiol., 2022).
We agree with the opinion that the paper: " Woo, S.L., Hermosa, R., Lorito M., Monte E. Trichoderma: a multipurpose, plant-beneficial microorganism for eco-sustainable agriculture. Nat. Rev. Microbiol. 2022, Nov 22, doi: 10.1038/s41579-022-00819-5." is the most up-to-date on this topic. As suggested, we introduced the citation of this work.
- In M&M section qPCR experiment design, authors specify how they prepare samples for standard curve construction. It would be helpful for readers if authors explain also test sample preparation from that 1mg of powder, the total gDNA concentration used for qPCR reactions, primers used and if an efficiency test/exclusion of dimmer formation has been performed in their work without needing to jump to reference 45!
Thanks to the Reviewer's comment, we understood that we had very briefly presented the stages of work carried out as part of the quantification of F. culmorum biomass. We rewrote this subsection. We tried to explain the method and purpose of preparing pure F. culmorum mycelium samples and mixed samples and their 10-fold dilution. We pointed out that this step was related to the determination of the specificity and accuracy of the method, and the DNA obtained from these samples was used for qPCR optimization.
We would also like to clarify that the references Nº 45 concerned the sequence of the primers used.
